# A simplified function-first method for the discovery and optimization of bispecific immune engaging antibodies

**Alex Shepherd[1,2], Bigitha Bennychen[1,2], Anne Marcil[1], Darin Bloemberg[1], Robert A. Pon[1], Risini D. Weeratna[1], Scott McComb[1,2,3]***

**1** Human Health Therapeutics Research Centre, National Research Council, Ottawa, Canada, **2** Department of Biochemistry, Microbiology and Immunology, University of Ottawa, Ottawa, Canada, **3** Centre for Infection, Immunity and Inflammation, University of Ottawa, Ottawa, Canada

\* scott.mccomb@nrc-cnrc.gc.ca

**Data Availability Statement:** All relevant data are within the paper and its Supporting information files.

## Abstract

Bi-specific T-cell engager antibodies (BiTEs) are synthetic fusion molecules that combine multiple antibody-binding domains to induce active contact between T-cells and antigen expressing cells in the body. Blinatumomab, a CD19-CD3 BiTE is now a widely used therapy for relapsed B-cell malignancies, and similar BiTE therapeutics have shown promise for treating various other forms of cancer. The current process for new BiTE development is time consuming and costly, requiring characterization of the individual antigen binding domains, followed by bi-specific design, protein production, purification, and eventually functional screening. Here, we sought to establish a more cost-efficient approach for generating novel BiTE sequences and assessing bioactivity through a function first approach without purification. We generate a plasmid with a bi-modular structure to allow high-throughput exchange of either binding arm, enabling rapid screening of novel tumour-targeting single chain variable (scFv) domains in combination with the well-characterized OKT3 scFv CD3-targeting domain. We also demonstrate two systems for high throughput functional screening of BiTE proteins based on Jurkat T cells (referred to as BiTE-J). Using BiTE-J we evaluate four EGFRvIII-scFv sequenced in BiTE format, identifying two constructs with superior activity for redirecting T-cells against the EGFRvIII-tumour specific antigen. We also confirm activity in primary T cells, where novel EGFRvIII-BiTEs induced T cell activation and antigen selective tumor killing. We finally demonstrate similar exchange the CD3-interacting element of our bi-modular plasmid. By testing several novel CD3-targeting scFv elements for activity in EGFRvIII-targeted BiTEs, we were able to identify highly active BiTE molecules with desirable functional activity for downstream development. In summary, BiTE-J presents a low cost, high-throughput method for the rapid assessment of novel BiTE molecules without the need for purification and quantification.

## Introduction

Monoclonal antibody (mAb) technology can be used to create biological molecules with high binding affinity and specificity for antigenic targets. In the case of immunomodulatory or

**Funding:** This research was funded by the National Research Council of Canada Human Health Therapeutics Research Centre.

**Competing interests:** No, authors have no conflicts of interest to declare.

cancer targeted therapeutics these antigens are typically on the surface of cells. Through binding, such antibodies can induce a variety of direct and indirect biological effects on the target proteins, such as agonist or antagonistic receptor modulation [1]. In addition to these direct effects, antibodies can also recruit immune cells through interactions with the antibody Fc domain, creating a connection between target cells and certain types of immune cells that can lead to phagocytosis or antibody-dependent cytotoxicity [1]. Synthetic immunology approaches have been developed to broaden the effects of antibodies and derivative molecules in immune cell activation, including the development of bi-specific T-cell engager (BiTE) antibodies that can induce strong antigen-targeted T cell responses [2]. The most clinically advanced of such therapies is Blinatumomab, a CD19xCD3 antibody [3] used in the treatment of acute lymphoblastic B-cell leukemia [4].

To further broaden the types of cancer that can be effectively targeted using BiTE therapeutics, optimize BiTE biological activity, and explore additional modalities for other types of bi-specific antibodies, there is a need for cost-effective high-throughput platforms for BiTE discovery and development. BiTE molecular development begins with the identification of two antigen-binding molecules, most typically single chain variable fragments (scFv) derived from monoclonal antibodies (mAb) isolated from mouse, human, or other animals. One scFv must bind an antigen on a target cell, with the other binding a T cell specific protein, with the most common T-cell target being CD3 [5] (see Fig 1A). These scFvs are connected using a linker, usually composed of a flexible amino acid sequence (GGGGS) length as seen in Blinatumomab [6], though this spacer element can be repeated or of varying composition to create a longer spacing between engager domains (GGGGS x 2, 3 etc.). In the presence of a BiTE molecule, T

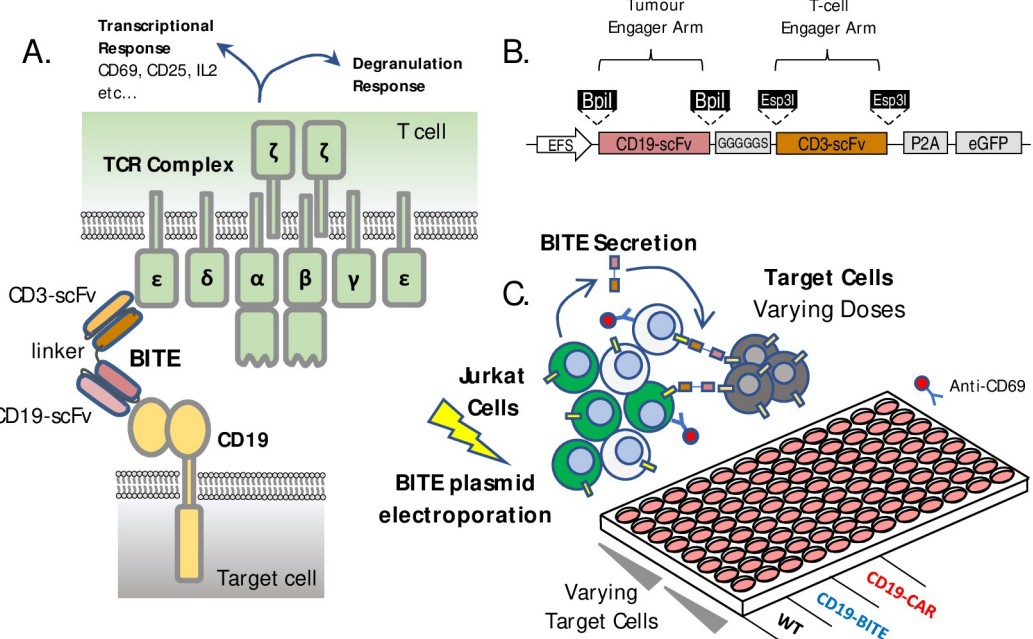

**Fig 1. Modular BiTE Plasmid and screening platform design.** (A) BiTE mechanism of action relies on simultaneous engagement of target cell receptors usually located on tumour cells and the CD3 domains of the T-cell receptor on T cells. This results in forming an active immune synapse between the T cell and the cancer, leading to both transcriptional and direct degranulation responses within the T cells, and target cell death. (B) The transgene design for a dual-modular BiTE plasmid incorporating type-IIs restriction enzyme cassettes on each scFv region to allow easy exchange of scFv domains via golden gate cloning, (C) BiTE-Jurkat screening assay setup uses direct electroporation of BiTE plasmids into Jurkat cells in co-culture with various target antigen positive or negative target cell lines.

cells and target cells expressing the target antigen should form a strong interaction, prompting both T cell activation and target cell killing [7,8].

Here we outline a complete, high-throughput method for generating novel bi-specific anti-body plasmids, rapidly producing BiTE proteins, and performing function first activity screening in the human Jurkat T cell line with no antibody preparation, purification or characterization. This allows for immediate functional insight for BiTE candidates following DNA assembly, reducing the time and cost associated with finding a lead candidate. Using the tools and techniques outlined in this paper, researchers can quickly discover and optimize ideal candidates for further clinical development or incorporation into more complex synthetic therapeutics.

## Materials and methods

### Construction of pBiTE modular plasmid

A synthetic plasmid modeled on blinatumomab [9] with modularized binding domains to allow low-cost single-pot restriction ligation recombination of either binding arm was designed in silico using A plasmid Editor [10]. The design contains a IgG1 signal peptide, modularized CD19-scFv (OKT3HD37), linker, modularized CD3-scFv (OKT3), and P2A-NeonGreen (see S1 Table for DNA and amino acid sequences used, see Fig 1B for a schematic of the sequence design). To build the pBiTE plasmid, CAR-specific domains were removed from pSLCAR-CD19 (Addgene #135992), leaving a linearized plasmid containing a short EF1a promoter and a P2A-NeonGreen reporter sequence within a linearized lenti-viral plasmid backbone. Next, a custom DNA fragment was synthesized (Twist Bioscience, USA) coding for the HD37 anti-CD19 scFV, linker, OKT3 anti-CD3 scFv and adapter sequences to allow cloning into the linearized plasmid using Gibson assembly [11]. Gibson assembly was performed using either the GeneArt™ Gibson Assembly HiFi Master Mix (ThermoFisher A46627) or an in house "DIY Gibson" assembly mixture using the RFC57 recipe [12]. Custom fragments (1uL), linearized backbone (1ul) and Gibson Assembly Mix (5ul) were combined and filled to 10ul with molecular grade water. The Gibson reaction was run in a thermocycler at 50˚C for 45 minutes before transformation. The reaction mix was then stored at -20˚C for an undefined period before proceeding to bacterial transformation.

The final plasmid incorporates modular restriction sites to allow swapping of the tumour engager arm (using BpiI) and the CD3 binding arm (using Esp3I), as well as the linker arm between the two (see Fig 1B). The constructed BiTE plasmid was transformed into DH5a chemically competent *E. coli*. To confirm successful cloning, individual transformed colonies were analyzed via direct colony PCR using backbone or scFv-specific primers. Colonies which contain predicted PCR products are grown overnight in normal lysogeny broth (LB) with 1% ampicillin. Plasmid DNA was purified using standard mini-plasmid or midi-plasmid purification kits (Cat#K0502 or #K0481, Thermo Fisher, USA) and then sequenced via Sanger sequencing to ensure accurate sequence construction. This plasmid, known as BMv4 can be found in the S1 Table t or on Addgene (Addgene #190677—Non His, or #190678—With His tag).

### Tumor antigen binding domain or CD3-binding domain exchange

When designing scFv inserts for "Golden-Gate" single-pot restriction/ligation [13], the 5' and 3' ends are structured depending on the cloning site used. Typiccally, scFv sequences were synthesized based on antibody heavy and light chain sequences to create a VH-linker-VL sequence that is approximately 600–800 bp long. For swapping of the N-terminal tumour-

antigen binding domain in pBITE the following BpiI restriction site containing sequence design was used: 5'–NNN**GAAGAC**NNAGGA—*VH or VL sequence—Linker—VH or VL sequence*—NNCCTTNN**GTCTTC**NNN-3'. For swapping of the C-terminal immune cell engaging domain using Esp3I the following sequence design was used 5'–NNN**CGTCTC**NAAGT—*VH or VL sequence—Linker—VH or VL sequence*—GCTAN**GAGACG**NNN-3'. Any internal Esp3I or BpiI sites in the scFv were altered with silent mutations. Specific primers we used for sub-cloning scFv sequences from previous EGFRvIII-CAR constructs can be found in S2 Table. In this manuscript a VH-Linker-VL orientation was used for all scFvs tested, though the reverse orientation has been show to produce functional scFv molecules as well [14]. Once assembled in silico, scFv sequences were generated through PCR subcloning or synthesized via commercial DNA synthesis company (Twist Bioscience, USA).

To swap either scFv, the following reaction conditions were used: 250ng of pBiTE, 50ng of new scFv DNA fragment, 0.25ul BpiI or Esp3I, 0.25ul T4 ligase, 4ul T4 reaction buffer and filled to 20ul with molecular grade $H_2O$. The scFv insertion reaction is run on a thermocycler under the following protocol: (37˚C for 10 min, 16˚C for 10 min) 10 cycles, 37˚C for 60 min, 80˚C for 5 min to inactivate enzymes, and 4˚C indefinitely. 2 to 5ul of this reaction was then transformed into chemically competent DH5α *E. coli*, which were then plated on ampicillin containing agar plates and grown overnight at 37˚C. To confirm successful cloning, individual transformed colonies were analyzed with PCR using backbone or scFv specific primers (see S2 Table for primers used). Colonies which contained predicted PCR products are grown overnight in normal lysogeny broth (LB) with 1% ampicillin. Plasmid DNA was then purified using standard mini- or midi-prep purification kits (Qiagen, USA). It is recommended that purified plasmids are sequenced to ensure correct construction, sequencing primers used for Sanger sequencing of the plasmid can we found in the S2 Table. EGFRvIII-specific scFv sequences used for cloning can be found in [15,16], and can be found in the S1 Table or on the Addgene repository (F263-4E11 Addgene #190680, F265-5B7 Addgene #190679, F269-3D12 Addgene #190681, and F271-1D2 Addgene #190682). Swapping CD3-targeting domains was performed similarly as described above using Esp3I enzyme to integrate synthetic CD3-scFv fragments (Twist Biosciences, USA), derived from in-house generated CD3-specific murine monoclonal antibodies, into the pBiTE construct. Novel CD3 monoclonal antibody and scFv sequences can be found in [17]. Plasmids encoding novel CD3 BiTEs can be obtained under MTA with the National Research Council Canada.

## Cell lines

All cell lines were monitored regularly for mycoplasma contamination using in-house PCR assay [18]. In preparation of cell assays, healthy cultures of Jurkat E6-1 cells (ATCC#TIB-152) were maintained in complete RPMI (RPMI1640 supplemented with 10% FBS, 2mM L-glutamine, 1mM sodium pyruvate and 100 μg/mL penicillin/streptomycin) with cell density between 0.25 and $1 \times 10^6$ cells/mL for several weeks; we have found that maintenance conditions are critical for consistent results in Jurkat activation assays. All target cell lines described in this paper were modified using Nuclight-Red Lentiviral reagent (Cat#4625, Sartorius, USA) to generate stable red-fluorescent cells which can be easily differentiated from effector cells in FACS or live microscopy analyses. Specific target lines used were as follows: Raji (CD19+, ATCC#CCL-86), Nalm6 (CD19+, a gift from Dr. Beat Bornhauser, Kinderspital Zürich, Switzerland), MCF7 (CD19-, ATCC#HTB-22), U87MG-vIII (EGFRvIII+, a gift from Prof. Cavnee, Ludwig Cancer Institute, USA), U87MG (EGFRvIII -, also a gift from Prof. Cavnee), DKMG (EGFRvIII^low, DSMZ#ACC277). Target cell lines were grown in varying media conditions, as recommended by cell repository.

## Jurkat direct BiTE electroporation protocol

Healthy cultures of Jurkat cells with cell density between 0.25 and $1\times10^6$ cells/mL were maintained in culture for several weeks. Prior to electroporation, a recovery plate was prepared using pre-warmed RPMI 1640 supplemented with 20% fetal bovine serum and 100 ug/ml of L-glutamate. Jurkat cells were pelleted via centrifugation, then resuspended in 100ul of 1SM buffer (as per [19]) and 2ug of plasmid and brought to room temperature. Cells were then placed into a 0.2cm electroporation cuvette (Cat#1652086, Bio-Rad, USA) and electroporated using a Lonza Nuceleofactor Electroporator using X4 settings. Cells were then immediately transferred to the recovery plate. Cells were allowed to recover for 4 hours prior to analysis or use in an assay. Post-recovery Jurkat cells and target cells were then resuspended in complete RPMI and placed in 96-well plates at varying ratios of Jurkat and target cells as shown in the text. If adherent target cells were used, cells were first treated with Accutase (Cat#A1110501, Thermo Fisher, USA) to create a single cell suspension. Plates were incubated at 37˚C, 5% $CO_2$ for either 16 to 40 hours. Plates was then stained with anti-CD69 APC (Cat#340560, BD Bioscience, USA) at 0.25ul per well in no-wash format. Plates were then incubated in the dark at RT for 15 minutes and analyzed via flow cytometry using a BD Fortessa device (BD Bioscience, USA).

## HEK293T transfection for BiTE supernatant production

HEK293T cells were plated in a 6 well plate at approximately 150k cell per well and allowed to grow overnight before transfection. The following transfection mixture was prepared: 1.2ml of serum free DMEM 200ul/well, 9ug/well linear polyethyleneimine (PEI; Cat#765090-1G, Sigma-Aldrich, USA), and 12ug (2ug/well) of pBiTE plasmid. The tube containing the transfection mix was vortexed and incubated at RT for 10 minutes. 200ul of the solution was added drop wise to each well incubated at RT for 20 minutes. Cells were then incubated at 37˚C for 4 hours. After 4 hours, media was removed and replaced with 2-3ml of fresh DMEM media supplemented with 10% Fetal Bovine Serum (FBS), 1% L-Glutamate (Cat#25030081, Gibco, USA) and 1% Penicillin-Streptomycin (Pen-Strep; Cat#15070063, Gibco, USA) (referred to as DMEM complete below). After 5–7 days, transfected HEK293T supernatant was removed and filtered using 0.45-micron filter to remove any excess cells. BiTE-containing supernatant was then frozen at -80˚C and thawed prior to BiTE-Jurkat assay as described below. We confirmed that freezing resulted in no loss of BiTE activity for the active BiTE molecules reported in this manuscript.

## Indirect HEK supernatant BiTE-Jurkat activation assay

If adherent target cells were used, target cells were first treated with accutase (Cat#A1110501, Thermo Fisher, USA) to create a single cell suspension. Jurkat cells and target cells were then resuspended in complete RPMI and placed in 96-well plates at varying ratios of Jurkat and target cells as shown in the text. Unless otherwise stated in the text, 50ul of BiTE containing HEK supernatant were added to appropriate wells. Complete media was then added to equalize plating volume. Plates were incubated at 37˚C, 5% $CO_2$ for either 16 or 40 hours, followed by staining with anti-CD69 APC (Cat#340560, BD Bioscience, USA) at 0.25ul per well in no-wash format. Plates were incubated in the dark at RT for 15 minutes and analyzed via flow cytometry using BC Fortessa device (BD Bioscience, USA).

## Primary human T cell culture preparation

To prepare T cells, healthy donor blood samples were obtained under appropriate safety and ethics approvals by the Ottawa Hospital Research Institute (Ottawa, Canada). Whole blood

was diluted 1:1 with Hank's balanced salt solution (HBSS) and PBMCs were isolated by Ficoll-Paque™ density gradient centrifugation, centrifuging for 20 min at $700 \times g$ without applying a brake. The PBMC interface was carefully removed by pipetting and was washed twice with HBSS by stepwise centrifugation for 15 min at $300 \times g$. PBMCs were resuspended and counted by mixing 1:1 with Cellometer ViaStain™ acridine orange/propidium iodide (AOPI) staining solution and counted using a Nexcelom Cellometer Auto 2000 (Nexcelom BioScience, Lawrence, Massachusetts, USA). T cells from were then activated with Miltenyi MACS GMP T cell TransAct™ CD3/CD28 beads and seeded $1 \times 10^6$ T cells/ml in serum-free StemCell Immunocult™-XF media (Cat#10981, StemCell Technologies, Vancouver, Canada) with 20U/ml clinical grade human IL-2 (Novartis). T cells were then polyclonally expanded for 7 to 10 days in culture before cryopreservation of aliquots using complete media + 10%DMSO.

### BiTE-induced target cell killing assessment via live fluorescence microscopy

Cryopreserved polyclonally expanded primary human T cells (frozen on day 7–14 post activation), were thawed on the day of the experiment and counted and assessed for viability using Nexcelom Cellometer Auto 2000 (Nexcelom BioScience, Lawrence, MA, USA). Human T cells and red-fluorescent target cells were then resuspended in Immunocult-XF media supplemented with 100U/mL IL2. Cells were then transferred to a 96-well culture plate at 10 000 T cells per well and 2000 target cells per well. Plates were then placed in an S3 Incucyte Live Cell Imagine system (Sartorius, USA) and incubated at 37˚C and 5% $CO_2$ for up to 140 hours. Images were taken every 2 to 4 hours. Target cell growth was tracked using automated assessment of red fluorescent area percentage using the accompanying Incucyte analysis software. Final data was assembled and analyzed using Graphpad Prism.

## Results

### Creating a modularized bispecific plasmid

Previously we reported on a Jurkat based platform for screening novel chimeric antigen receptor molecules [20], here we wished to create a similarly flexible plasmid which could be used for functional screening of single-chain variable fragments (scFv) in a soluble bi-specific antibody molecules (Fig 1A). We based our plasmid design on the amino acid sequence for blinatumomab [9], adding type-IIs restriction sites to allow for simplified cloning of either binding arm of the molecule (Fig 1B). The modularized blinatumomab biosimilar plasmid (pBiTE) allows either end of the BiTE DNA to be swapped for an alternative scFv in a single pot restriction ligation reaction (Fig 1B), as well as exchange of the linker connecting both scFv domains if desired. This system can be used to easily customize the BiTE design, allowing for efficient construction of a variety of BiTEs and BiTE compositions for testing. Our testing shows highly efficient exchange of both target antigen scFv and T cell antigen scFv can be achieved consistently using single pot restriction ligation as described in the methods section.

### BiTE-J function first screening platform

Using the pBiTE plasmid, we first tested a functional screening method using Jurkat electroporation to induce transient expression and secretion of the BiTE molecule, similar to the method we have previously reported for screening novel chimeric antigen receptors [20]. Following BiTE plasmid electroporation, Jurkat cells were co-incubated with target cells at varying effector to target ratios to produce a standard dose response curve, with Jurkat cells acting as both the BiTE producer and immune effector cell (Fig 1C). After co-incubation of Jurkat

and target cells, cultures were stained with anti-CD69 and analyzed via flow cytometry to assess BiTE activity. While we were able to detect consistent increases in Jurkat activation with BiTE expression via this method (S1A Fig), activation of Jurkat cells expressing BiTE was lower than that observed for Jurkat cells expressing a CD19-targeted CAR (S1B Fig). We hypothesized that the low activity of the BiTE in this assay format might be due to relatively low electroporation efficiency and BiTE production in Jurkat cells, thus we wished to examine a more traditional transient production of BiTE proteins in HEK293T cells.

As a strategy to try to improve the signal to noise ratio for our BITE activity assay we next developed a transient BITE production method using HEK293T cells, which are widely used for production of recombinant proteins where a human cell source is desirable [21]. HEK293T cells were transfected with BiTE plasmids using standard PEI transfection. Fluorescence microscopy confirms successful transfection due to expression of the pBiTE GFP marker at day 1. HEK293T cells were then grown for 5–7 days before collecting the supernatant, using filtration to clear any remaining cells. Non-purified BiTE supernatants were then assayed using co-cultures of Jurkat T cells and target cells similarly to the previous iteration of the BiTE-J assay (Fig 2A). These co-cultures were left for either 16 or 40 hours, then analyzed via flow cytometry for CD69 upregulation on Jurkat cells. With this improved BiTE production technique, we observed more consistent BiTE-induced upregulation of CD69 in the presence of increasing numbers of CD19+ Raji or NALM6 target cells (Fig 2B). The addition of non-transfected HEK293T supernatant as a control had no effect on Jurkat activation.

As this production technique yielded a large volume of BiTE supernatant, much more than necessary for Jurkat experiments, we were also able to rapidly proceed to testing the effects of CD19-BiTE on co-cultures of primary human T cells and CD19+ Raji or Ramos target cells. Using live fluorescence microscopy (Incucyte), we monitored the relative growth of mKate2-labelled target cells in the presence of varying doses of CD19-BiTE supernatant. We observed strong dose-dependent target cell killing with BiTE treatment for both target cell types (Fig 2C). These results demonstrate that transient production in HEK293T cells followed by BiTE-J assay represents a viable platform for testing of biological activity for BiTE molecules.

## Swapping pBiTE tumor engager arm

Next, we wished to test the flexibility of our BiTE functional screening approach for different tumour antigen targeting domains. Using several mouse-derived single chain variable fragments (scFvs) targeting EGFRvIII, which we have previously reported to induce robust activity against EGFRvIII+ glioblastoma cells when combined in CAR proteins [20]. We thus generated novel EGFRvIII-BiTE constructs using single pot restriction-ligation cloning with pBiTE (Fig 3A). We then transiently transfected HEK293T cells with EGFRvIII-BiTE plasmids and screened supernatants via Jurkat co-culture with EGFRvIII-positive U87vIII and DKMG glioblastoma target cell lines. Only two of the BiTE plasmids generated (F269 and F263) resulted in positive CD69 upregulation when BiTE supernatants were added to Jurkat/target cell co-cultures (Fig 3B). We then proceeded to testing using polyclonally expanded primary human T cells in co-culture with target antigen-positive U87-vIII target cells, or target antigen-negative U87-WT or Raji target cells. In this case, we observed rapid killing of EGFRvIII-positive target cells (Fig 3C) but not EGFRvIII-negative cells (Fig 3D and 3E). Overall, these results demonstrate that our modular plasmid platform allows rapid cloning and functional screening of novel BiTE molecules for T-cell redirecting activity, identifying two candidate BiTE molecules F263-OKT3 and F269-OKT3 for potential future development.

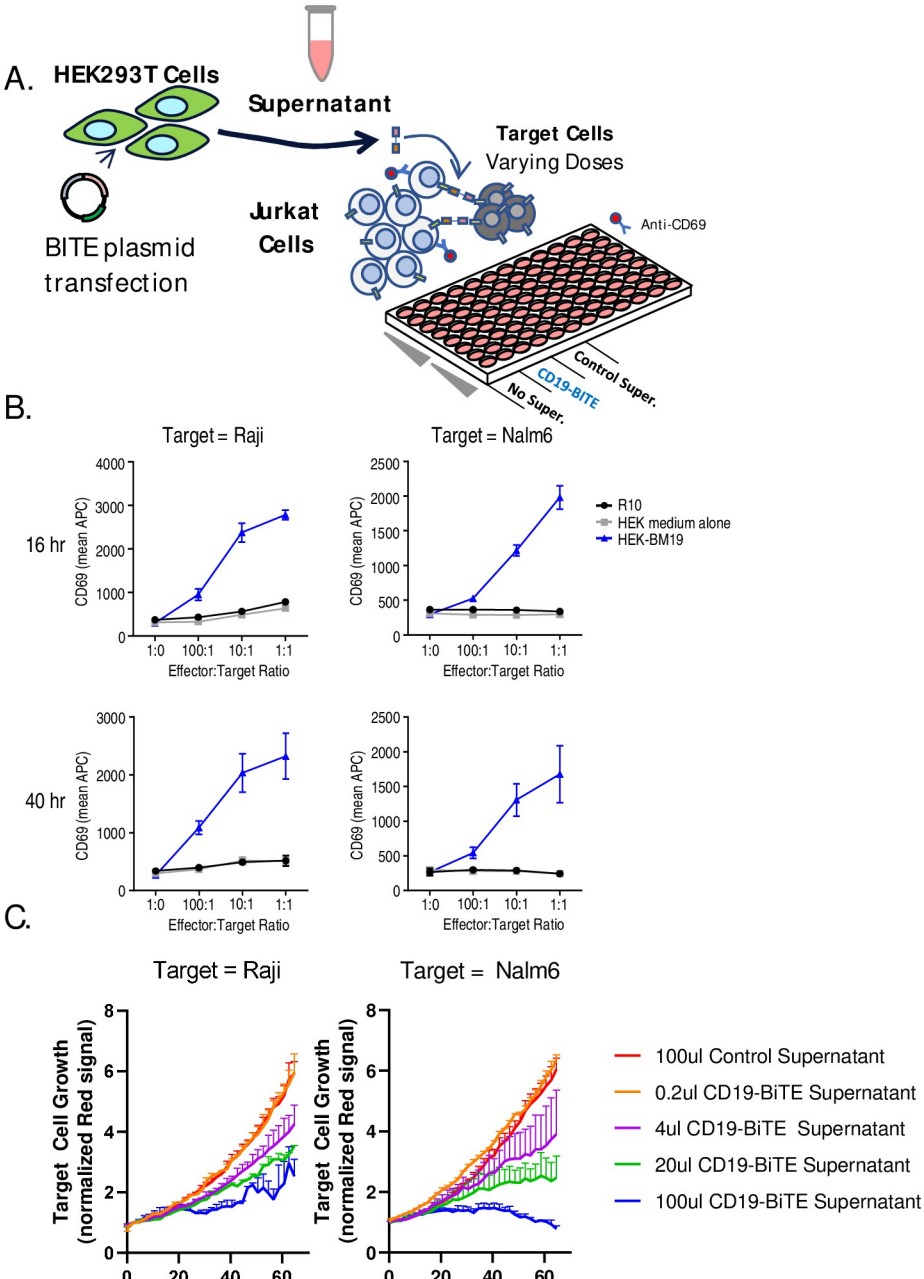

**Fig 2. BiTE-Jurkat assay using rapid HEK293T BiTE production.** (A) A schematic illustration of the workflow for production of BiTE supernatant using HEK293T cells and testing using Jurkat-target cell co-cultures is shown (B) HEK293T cells were transiently transfected with BiTE plasmid as described in the methods section and supernatants were collected and frozen at -80˚C. 50 000 Jurkat cells were placed in a 96-well plate in co-culture with varying numbers of CD19+ Raji or NALM6 with effector to target ratios as shown. 50ul of BiTE supernatant or control supernatant was then added and co-cultures were incubated at 37˚C overnight. Co-cultures were stained with anti-human CD69-APC and analyzed via flow cytometry. Graphs show the mean result from 4 experiments repeated in duplicate +/- SEM. (C) Primary human T cells were placed in co-culture with 10 000 human T cells and equal numbers of red fluorescently labelled CD19+ Raji or NALM6 cells, and varying doses of CD19-BiTE supernatant was added. Co-cultures were then monitored at regular intervals using fluorescence microscopy and automated cell counting via Incucyte. Graphs show the growth rate of red fluorescent target cells with varying BiTE dose. Primary T cell results are from a single experiment performed in duplicate.

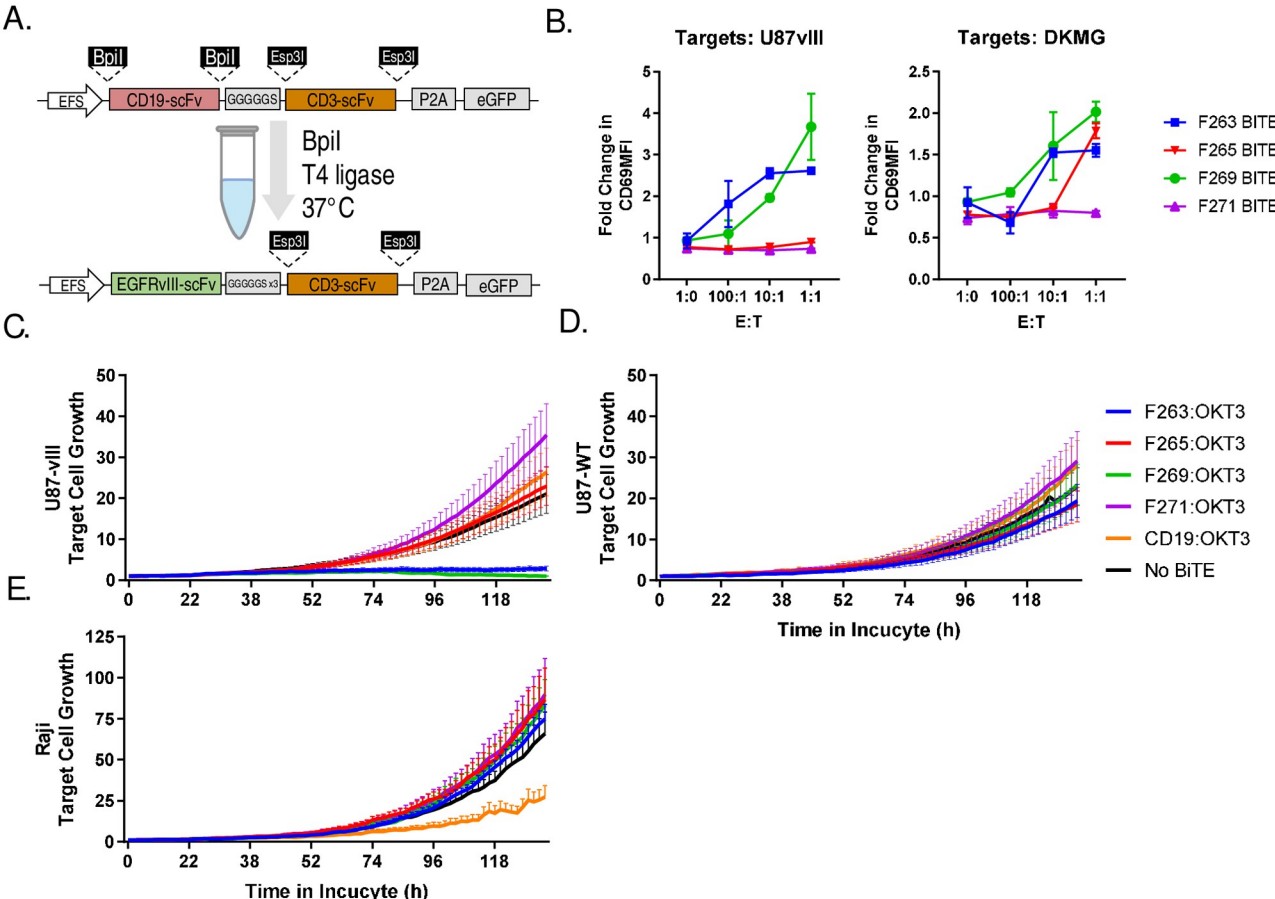

**Fig 3. Production and screening of EGFRvIII targeted BiTEs.** (A) A single-pot restriction ligation reaction with BpiI restriction enzyme was used to swap CD19-targeting arm with EGFRvIII-specific scFv sequences to generate EGFRvIII-BiTE plasmids as described in the methods section. (B) HEK293T cells were transiently transfected with BiTE plasmids and supernatant was collected and frozen. 50 000 Jurkat cells were placed in a 96-well plate in co-culture with varying numbers of EGFRvIII+ U87vIII or DKMG target cells as shown. 50ul of BiTE supernatant or control supernatant was then added and co-cultures were incubated at 37˚C overnight. Co-cultures were stained with anti-human CD69-APC and analyzed via flow cytometry. Graphs show the mean result from a single experiment performed in duplicate +/- SEM. Fresh EGFRvIII or CD19-specific BiTE supernatants were then generated in HEK293T cells for testing in primary T cells. 10 000 human T cells were combined with 2000 (C) EGFRvIII+ CD19- U87vIII target cells, (D) EGFRvIII- CD19- U87WT cells, or (E) EGFRvIII- CD19+ Raji cells. Graphs show the relative fluorescent signal of red-fluorescent target cells over 5 days in co-culture. Results are derived from a single experiment, but are representative of at least 3 repeated observations.

## Swapping the CD3 targeting domain

We next sought to test our platform for flexibility with respect to screening novel T-cell engaging domains of the bispecific molecule. We developed a number of novel mouse monoclonal antibodies (mAbs) against human CD3 complex using a multi-antigen immunization strategy in mice and traditional hybridoma screening. Through this process, we were able to identify several monoclonal antibodies with reactivity to Jurkat cells (Fig 4A) and human T cells (Fig 4B). To assess whether scFvs derived from novel CD3-targeted mAbs would be functional within as part of a BiTE molecule, we cloned 4 anti-human CD3 single chain variable fragments into CD19 or EGFRvIII specific BiTE plasmids (Fig 4C). We then generated supernatants using transient transfection of HEK293T as described above. BiTE supernatants were screened for activity using Jurkat cells in co-culture with EGFRvIII-expressing U87-VIII targets or CD19-expressing Raji cells. Previously tested constructs using OKT3 CD3-engager

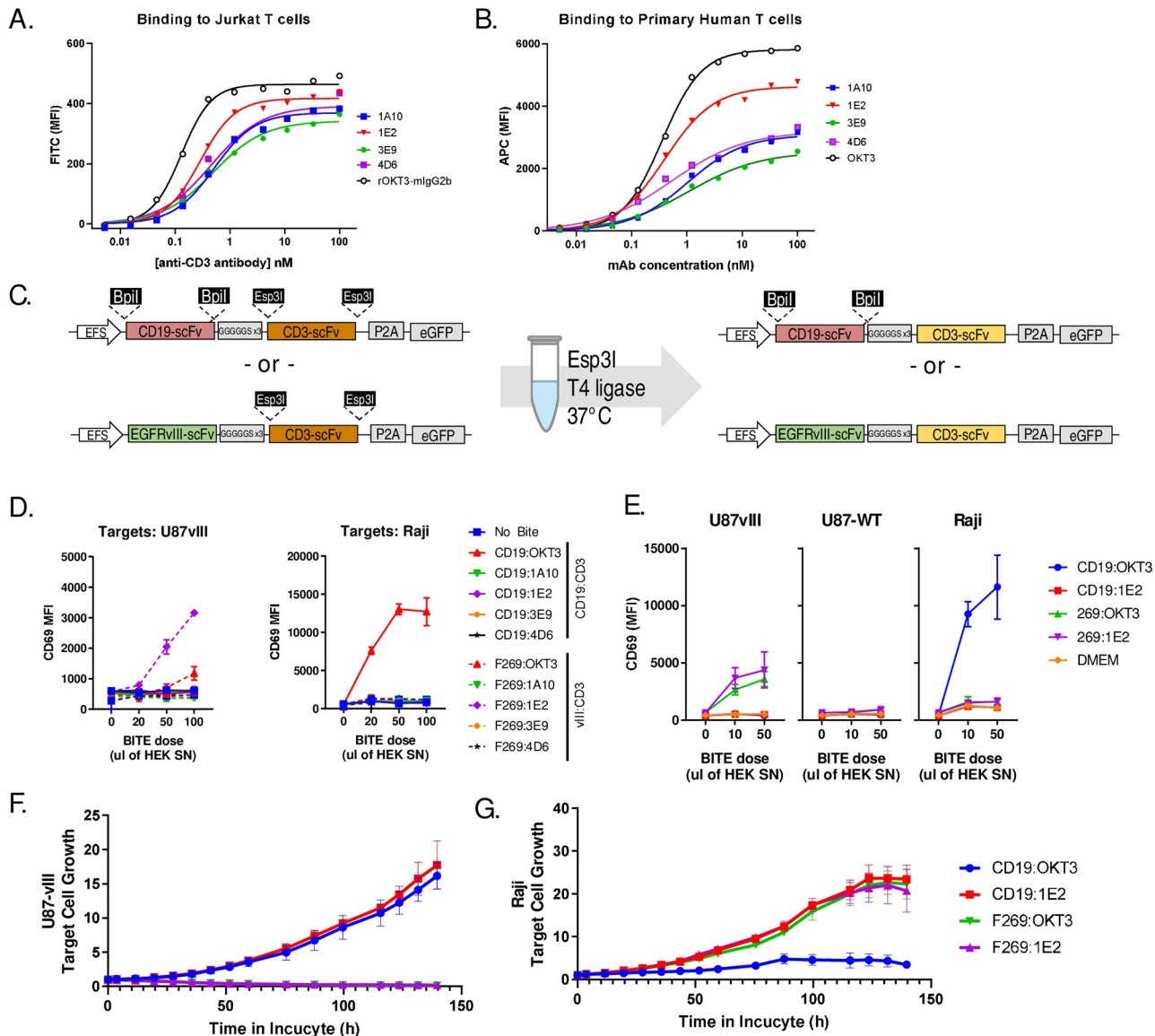

**Fig 4. Production and screening of BiTE molecules incorporating novel CD3 targeting scFvs.** Novel monoclonal antibodies were generated via mouse immunization and hybridoma screening. Four monoclonal antibodies are shown which have high reactivity to (A) Jurkat T cells and (B) primary human T cells. Following this, antibody heavy and light chains were sequenced. (C) scFv DNA was then synthesized and assembled into CD19 or EGFRvIII BiTE molecules using single-pot restriction ligation with Esp3I restriction enzyme. (D) Ten unique CD19 or EGFRvIII BiTE plasmids were then transfected into HEK293T cells and supernatants were tested using Jurkat cells in 1:1 co-cultures with EGFR-vIII+ U87vIII cells (*left*) or CD19+ Raji cells (*right*) using varying doses of BiTE supernatant as shown. (E) Four BiTE plasmids found to produce active BiTE molecules were tested again against EGFRvIII+ U87-vIII cells, EGFRvIII- U87-WT cells, or CD19+ Raji cells. (F) BiTE supernatants for CD19 or EGFRvIII targeted molecules incorporating the 1E2 CD3-scFv were added to co-cultures with 10 000 primary human T cells and 2000 target cells. Graphs show the relative fluorescent signal of red-fluorescent target cells over 5 days in co-culture. Results are derived from a single experiment, but are representative of at least 3 repeated observations.

arms showed activity with both EGFRvIII and CD19 specific BiTEs, whereas we detected activity for only one of our novel CD3-engager BiTEs and only when combined with an EGFRvIII-specific scFv (Fig 4D). To confirm these results, we repeated BiTE production and Jurkat co-culture screening of CD19 and EGFRvIII targeted BiTE molecules incorporating OKT3 or the novel 1E2 CD3-targeting single chain variable fragment. Whereas EGFRvIII BiTEs

incorporating an scFv derived from the 1E2 mAb or OKT3 showed specific reactivity against EGFRvIII expressing U87-vIII cells, only CD19-OKT3 showed reactivity to CD19-expressing Raji cells (Fig 4E). These results indicate that the novel CD3-targeting 1E2-scFv is functional only for an EGFRvIII-targeting BiTE but not a CD19-targeting BiTE, likely due to the specific binding characteristics of the CD19 or EGFRvIII scFv elements.

Finally, to assess whether the wholly novel EGFRvIII-1E2 BiTE molecule identified here can induce genuine functional interaction between human T cells and target cells, we proceeded to screening these molecules for activity in co-cultures of polyclonally expanded human T cells with U87-VIII or Raji target cells. Human T-cells quickly killed EGFRvIII+ target cells when treated with a BiTE combining EGFRvIII-scFv with OKT3 or our novel CD3-specific 1E2 scFv, but not CD19-targeted BiTEs (Fig 4F, S1–S4 Videos). In contrast, a CD19-specific scFv in combination with only OKT3 showed activity against Raji cells, but not a similar CD19-1E2 or EGFRvIII-specific BiTEs (Fig 4G, S5–S8 Videos). Overall, these results indicate that we have established a flexible platform for high-throughput functional screening of novel BiTE molecules wherein either the tumour targeting or immune targeting arm can be recombined and screened for biological activity.

## Discussion

Monoclonal antibody therapies targeting many cancer-associated antigens can effectively induce antibody dependent cellular cytotoxicity (ADCC) through simultaneous engagement of target antigens via their variable domains and immune cells via the Fc domain. Many recombinant antibodies have been shown to mediate their activity through ADCC both in pre-clinical and clinical studies of CD20-targeting rituximab [22], Her2-targeting trastuzumab [23], or CD38 targeting daratumumab [24], among others. While these therapies have had remarkable success in treatment of many cancers, NK-mediated ADCC lacks the aggressively proliferative, tumour-penetrating, and inflammatory responses that can be mediated by antigen-specific T cell responses. Thus, bi-specific T cell engaging antibodies were envisioned as a means of redirecting more potent T cell responses against tumours, leading to the development of blinatumomab, a CD19-CD3 BiTE used to treat acute lymphoblastic leukemia, as well as other BiTE therapeutics at various stages of pre-clinical and clinical development [6]. While BiTE therapeutics show strong potential for treatment of many different cancer types, their complex structure makes discovery and development costly and unpredictable. Thus, we sought to establish a flexible and cost-efficient platform for rapid identification of promising BiTE molecules.

The system presented here provides a full overview of our development work for BiTE screening, including: design of a bi-modularized BiTE plasmid, scFv swapping of either tumor or T-cell engaging elements, transient production in HEK293T cells, and BiTE-J screening of supernatants. This establishes a complete method for high throughput functional screening of novel BiTE molecules which can be readily replicated in other labs interested to develop their own BiTEs. Other systems have been previously reported for BiTE screening, but all are dependent on production of purified high-quality proteins; such as the dock and lock system used by Rossi et al. to screen multi-functional antibodies [25]; the work of Zappala et al. where antibodies are covalently linked to a CD3-specific scFv [26]; or that by Hofmann et al. [27] wherein intein mediated dimerization of antibodies is used. Whereas these methods offer a compelling means to mix and match many different binding domains for their activity in dimerized format, all require purified antibody proteins, which may increase the complexity and cost of screening. Furthermore, some methods may not be predictive of activity for BiTEs incorporated in a single molecule for downstream production. It should be noted that while

the BiTE-J method presented here does not require a purification step to screen and evaluate generated BiTEs, purification will ultimately be necessary for downstream testing using *in vivo* models. Given that the plasmid we developed here does not code for an Fc region, similar to blinatumomab, making standard protein G purification not possible. Thus, we intend to investigate alternate versions of this platform in future, for example we have developed a version of this BiTE screening plasmid containing a His-tag to allow downstream purification via IMAC column.

Sugiyama et al have demonstrated a similar workflow wherein more than 52 individual bi-specific antibody molecules were generated and screened in both heavy/light chain orientations [28]. We propose that the plasmids established here provide a more flexible cloning system, as the application of single-pot restriction ligation cloning would allow users with limited molecular biology knowledge to generate novel BiTE assemblies via golden gate cloning, while also minimizing the synthesis costs for labs employing these tools. Downstream of bi-specific antibody generation and purification, most previous reports have focused on some form of T-cell Dependent Cellular Cytotoxicity (TDCC) assay [29], typically using a viability measurements such as MTS or Luciferase reporting systems. These assays can be effective; however, we find that assessment of CD69 on Jurkat cells to be the most easily scalable method of measuring T cell activation activity. Jurkat testing also removes the need for donor cells and provides a consistent, standardized cell line for screening BiTEs. For testing with primary T cells, we find that live microscopy provides a simple method of evaluating active tumor control and killing, although flow cytometry can also be applied to yield similar results.

Other more advanced and higher throughput approaches to screening and analysis of bi-specific antibodies based on single cell droplet based microfluidics fluorescence sorting have also been reported [30,31]. As such approaches combine both antibody panning and antibody activation into a singular assay, they are compatible with the screening of polyclonal libraries of bi-specific molecules. The plasmids we have developed here also incorporate elements necessary for lentiviral production, and thus would be compatible with the generation of polyclonal Jurkat-BiTE cell libraries. In future, we will investigate suitable methods for downstream functional assessment such as single cell sorting and screening, or microwell entrapment of BiTE producing and target cells.

While other methods of BiTE screening such as those discussed above are have been developed, many of these methods require significant investment and access to specialized equipment and the use of costly consumables. BiTE-J was designed not only to allow for quick BiTE discovery and optimization, but also to provide tools and protocols to academic labs, which can be performed with minimal protein production and purification costs, and with minimal DNA synthesis costs. We hope that the ease of access and relatively low entrance cost of BiTE-J allows more researchers to contribute to the development and optimization of bi-specific therapeutics, the development of novel synthetic biology applications for BiTE molecules, and to apply these tools more broadly to biological research.

This work also extends our insights into the specific EGFRvIII-specific antibodies tested here, which we have previously reported on for CAR activity [20]. The four EGFRvIII-specific scFv molecules tested all showed some activity in CAR format, both in the activation of CAR-expressing Jurkat cells or for inducing primary CAR-T killing of U87-VIII target cells. In contrast to this, we find that only two molecules showed significant activity here when tested in BiTE format. As the screening approach employed here does not incorporate any characterization of BiTE production, it is possible that low biological activity for these molecules may have been due to low productivity within HEK293T cells, and thus we cannot state conclusively that these molecules would not be functional if produced and purified in other formats. While this

may be the case, low production yield is also an important category for considering whether BiTE molecules are suitable for therapeutic development and thus would need to be considered in molecular hit selection regardless.

Similarly, we also find that only one of the novel CD3-targeting scFvs tested here showed activity in BiTE format, and even then, activity was restricted to combination with EGFRvIII-targeting scFv and not with a CD19-specific scFv. Again, we have no insight into the failure rate for BiTE molecules in this assay, as the intent of this assay is to provide a rapid means for testing biological activity rather than focusing on various aspects of antibody characterization. We are currently undertaking molecular studies to test whether incorporating different linker domains may be able to improve the activity of BiTEs using the novel CD3-scFv reported to have activity here, something that may provide further insight into the design constraints for these BiTE molecules.

The fully novel EGFRvIII-CD3 engaging molecules identified at the end of the manuscript provides a proof of principle example for high throughput discovery of novel BiTE therapeutics. As this molecule shows strong activity in mediating T-cell killing of antigen-positive tumour cells and little to no off-target activity, it also shows potential for downstream development as a therapeutic to target EGFRvIII-expressing tumours, such as glioblastoma. In the future, we intend to expand on this platform to better understand the design space and study the biology of bi-specific antibody therapeutics that mediate active engagement of T-cells or other immune cells with target cells.

## Supporting information

**S1 Fig. BITE-Jurkat Assay using direct electroporation in Jurkats.** (A) Human Jurkat T cells were electroporated with CD19-targeted BITE or CAR plasmids as described in the methods section. Following recovery, electroporated Jurkat cells were then placed in co-culture with fluorescently labelled CD19-expressing target cells at various effector to target ratios and incubated at 37˚C overnight. Co-cultures were stained with anti-human CD69-APC and analyzed via flow cytometry. Results are representative of 3 repeated experiments. (B) The mean fluorescent intensity for CD69-APC staining on gated Jurkat cells is shown for Jurkat-CD19-BITE or Jurkat-CD19-CAR cells in co-culture with CD19+ Raji or NALM6 cells for 18 hours (top) or 42 hours (bottom). Graphs show the mean result from 3 experiments repeated in duplicate +/- standard error of the mean (SEM).
(TIF)

**S1 Table. Specific DNA and amino acid sequences for the constructs used in this study are provided.**
(XLSX)

**S2 Table. Primer sequences used in this manuscript are provided.**
(XLSX)

**S1 Video. Primary T cells show no response to U87vIII target cells with CD19-OKT3 BITE.** BITE supernatant for CD19 specific BITE (CD19-OKT3) was generated using HEK293T cell as described in the text, raw BITE supernatant was then added to co-cultures with 10 000 primary human T cells (unlabelled) and 2000 target cells (stable nuclear localized red fluorescent protein). Phase contrast and red fluorescent images were acquired hourly using in incubator live fluorescence microscopy device (Incucyte), and combined in overlay to generate video file. Results are representative of at least 3 replicates.
(MP4)

**S2 Video. Primary T cells show no response to U87vIII target cells with CD19-1E2 BITE.**
BITE supernatant for CD19 specific BITE (CD19-1E2) was generated using HEK293T cell as
described in the text, raw BITE supernatant was then added to co-cultures with 10 000 primary
human T cells (unlabelled) and 2000 target cells (stable nuclear localized red fluorescent pro-
tein). Phase contrast and red fluorescent images were acquired hourly using in incubator live
fluorescence microscopy device (Incucyte), and combined in overlay to generate video file.
Results are representative of at least 3 replicates.
(MP4)

**S3 Video. Primary T cells show immediate killing of U87vIII target cells with F269-OKT3
BITE.** BITE supernatant for EGFRvIII specific BITE (F269-OKT3) was generated using
HEK293T cell as described in the text, raw BITE supernatant was then added to co-cultures
with 10 000 primary human T cells (unlabelled) and 2000 target cells (stable nuclear localized
red fluorescent protein). Phase contrast and red fluorescent images were acquired hourly
using in incubator live fluorescence microscopy device (Incucyte), and combined in overlay to
generate video file. Results are representative of at least 3 replicates.
(MP4)

**S4 Video. Primary T cells show immediate killing of U87vIII target cells with F269-1E2
BITE.** BITE supernatant for EGFRvIII specific BITE (F269-1E2) was generated using
HEK293T cell as described in the text, raw BITE supernatant was then added to co-cultures
with 10 000 primary human T cells (unlabelled) and 2000 target cells (stable nuclear localized
red fluorescent protein). Phase contrast and red fluorescent images were acquired hourly
using in incubator live fluorescence microscopy device (Incucyte), and combined in overlay to
generate video file. Results are representative of at least 3 replicates.
(MP4)

**S5 Video. Primary T cells show immediate killing of Raji target cells with CD19-OKT3
BITE.** BITE supernatant for CD19 specific BITE (CD19-OKT3) was generated using
HEK293T cell as described in the text, raw BITE supernatant was then added to co-cultures
with 10 000 primary human T cells (unlabelled) and 2000 target cells (stable nuclear localized
red fluorescent protein). Phase contrast and red fluorescent images were acquired hourly
using in incubator live fluorescence microscopy device (Incucyte), and combined in overlay to
generate video file. Results are representative of at least 3 replicates.
(MP4)

**S6 Video. Primary T cells show no response to Raji target cells with CD19-1E2 BITE.** BITE
supernatant for CD19 specific BITE (CD19-1E2) was generated using HEK293T cell as
described in the text, raw BITE supernatant was then added to co-cultures with 10 000 primary
human T cells (unlabelled) and 2000 target cells (stable nuclear localized red fluorescent pro-
tein). Phase contrast and red fluorescent images were acquired hourly using in incubator live
fluorescence microscopy device (Incucyte), and combined in overlay to generate video file.
Results are representative of at least 3 replicates.
(MP4)

**S7 Video. Primary T cells show no response to Raji target cells with F269-OKT3 BITE.**
BITE supernatant for EGFRvIII specific BITE (F269-OKT3) was generated using HEK293T
cell as described in the text, raw BITE supernatant was then added to co-cultures with 10 000
primary human T cells (unlabelled) and 2000 target cells (stable nuclear localized red fluores-
cent protein). Phase contrast and red fluorescent images were acquired hourly using in incuba-
tor live fluorescence microscopy device (Incucyte), and combined in overlay to generate video

file. Results are representative of at least 3 replicates.
(MP4)

**S8 Video. Primary T cells show no response to Raji target cells with F269-1E2 BITE.** BITE supernatant for EGFRvIII specific BITE (F269-1E2) was generated using HEK293T cell as described in the text, raw BITE supernatant was then added to co-cultures with 10 000 primary human T cells (unlabelled) and 2000 target cells (stable nuclear localized red fluorescent protein). Phase contrast and red fluorescent images were acquired hourly using in incubator live fluorescence microscopy device (Incucyte), and combined in overlay to generate video file. Results are representative of at least 3 replicates.
(MP4)

## Author Contributions

**Conceptualization:** Anne Marcil, Darin Bloemberg, Scott McComb.

**Formal analysis:** Alex Shepherd, Bigitha Bennychen, Anne Marcil, Darin Bloemberg, Robert A. Pon, Risini D. Weeratna, Scott McComb.

**Funding acquisition:** Scott McComb.

**Investigation:** Alex Shepherd, Bigitha Bennychen, Anne Marcil, Darin Bloemberg, Robert A. Pon, Scott McComb.

**Methodology:** Alex Shepherd, Bigitha Bennychen, Darin Bloemberg, Robert A. Pon, Scott McComb.

**Project administration:** Anne Marcil, Darin Bloemberg, Risini D. Weeratna, Scott McComb.

**Resources:** Anne Marcil, Scott McComb.

**Supervision:** Anne Marcil, Risini D. Weeratna, Scott McComb.

**Visualization:** Scott McComb.

**Writing – original draft:** Alex Shepherd, Risini D. Weeratna, Scott McComb.

**Writing – review & editing:** Bigitha Bennychen, Anne Marcil, Darin Bloemberg, Robert A. Pon, Risini D. Weeratna, Scott McComb.

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
