## [Editor Report · Decision Letter 0]

27 Sep 2022

PONE-D-22-22870A Simplified Function-First Method for the Discovery and Optimization of Bispecific Immune Engaging AntibodiesPLOS ONE

Dear Dr. McComb,

Thank you for submitting your manuscript to PLOS ONE. After careful consideration, we feel that it has merit but does not fully meet PLOS ONE’s publication criteria as it currently stands. Therefore, we invite you to submit a revised version of the manuscript that addresses the points raised during the review process.

We look forward to receiving your revised manuscript.

Kind regards,

Brahma Nand Singh

Academic Editor

PLOS ONE

Journal Requirements:

Additional Editor Comments:

Authors have manuscript presented the development of a screening platform for the discovery of novel bi-specific T-cell engager molecules. Overall research are interesting and impactful.

However, authors should be improve the methods and discussion parts of the paper.

---

## [Author Response · Author response to Decision Letter 0]

24 Nov 2022

We have revised the methods and discussion sections as well as edited the entire manuscript. We have also now deposited the key plasmid reagents generated here at the Addgene repository and will release for the community to be able to access them at the time of publication. We look forward to a positive reception for the revised manuscript.

---

## [Decision Letter · Decision Letter 1]

17 Feb 2023

PONE-D-22-22870R1A Simplified Function-First Method for the Discovery and Optimization of Bispecific Immune Engaging AntibodiesPLOS ONE

Dear Dr. McComb,

Thank you for submitting your manuscript to PLOS ONE. After careful consideration, we feel that it has merit but does not fully meet PLOS ONE’s publication criteria as it currently stands. Therefore, we invite you to submit a revised version of the manuscript that addresses the points raised during the review process.

We look forward to receiving your revised manuscript.

Kind regards,

Brahma Nand Singh

Academic Editor

PLOS ONE

Reviewers' comments:

Reviewer's Responses to Questions

**Comments to the Author**

1. If the authors have adequately addressed your comments raised in a previous round of review and you feel that this manuscript is now acceptable for publication, you may indicate that here to bypass the “Comments to the Author” section, enter your conflict of interest statement in the “Confidential to Editor” section, and submit your "Accept" recommendation.

Reviewer #1: (No Response)

2. Is the manuscript technically sound, and do the data support the conclusions?

Reviewer #1: Yes

3. Has the statistical analysis been performed appropriately and rigorously? 

Reviewer #1: Yes

4. Have the authors made all data underlying the findings in their manuscript fully available?

Reviewer #1: Yes

5. Is the manuscript presented in an intelligible fashion and written in standard English?

Reviewer #1: Yes

6. Review Comments to the Author

Reviewer #1: In the mansucript ‘A Simplified Function-First Method for the Discovery and Optimization of Bispecific Immune Engaging Antibodies’, Shepherd et al. establish a cost effective approach for generating novel BiTE sequences and assessing functional bioactivity bypassing the need for purification. They have generated a bi-modular plasmid and established BiTE-J platform for functional screening. The study is informative, and the findings are relevant for the readership of PLOS ONE.' The study design and experimental approaches are appropriate, and the data generally supports the conclusion. The authors need to address some conceptual and technical concerns in the current version of the manuscript before its publication. The issues are listed below.

Major:

The data in Figure 2, in its present form, should be moved to supplementary.

The authors state that ‘The addition of non-transfected HEK293T supernatant as a control had no effect on Jurkat activation.’ It would be helpful to include this data.

Although the authors have quantified T cell activation and Target Cell Growth, have the authors considered assessing the cytotoxicity of T cells such as by quantifying the Tumor cell lysis and T cell mediated cytotoxicity?

It is not immediately clear what E:T ratios were used for experiments in Figure 5.

Minor:

Please fix line 83. ‘candidate. .’

7. PLOS authors have the option to publish the peer review history of their article (what does this mean?). If published, this will include your full peer review and any attached files.

Reviewer #1: No

---

## [Author Response · Author response to Decision Letter 1]

10 Mar 2023

Line by line response to reviewers provided below. Document version is also attached with manuscript files. 

Reviewer #1: In the mansucript ‘A Simplified Function-First Method for the Discovery and Optimization of Bispecific Immune Engaging Antibodies’, Shepherd et al. establish a cost effective approach for generating novel BiTE sequences and assessing functional bioactivity bypassing the need for purification. They have generated a bi-modular plasmid and established BiTE-J platform for functional screening. The study is informative, and the findings are relevant for the readership of PLOS ONE.' The study design and experimental approaches are appropriate, and the data generally supports the conclusion. The authors need to address some conceptual and technical concerns in the current version of the manuscript before its publication. The issues are listed below.

We appreciate Reviewer 1s comments and belief that this manuscript meets the standards for PLOS ONE publication. We have endeavoured to address the remaining concern listed here 

The data in Figure 2, in its present form, should be moved to supplementary.

We agree that the data shown in what was previously Figure 2, while important, is not the central focus of the text. The figure has been moved to supplementary and the paper has been changed to reflect that. 

The authors state that ‘The addition of non-transfected HEK293T supernatant as a control had no effect on Jurkat activation.’ It would be helpful to include this data.

Non-transfected HEK293T media is now included as a control in Figure 2B (Previously 3B). This data should be sufficient to show the lack of effect from transfected HEK293T media 

Although the authors have quantified T cell activation and Target Cell Growth, have the authors considered assessing the cytotoxicity of T cells such as by quantifying the Tumor cell lysis and T cell mediated cytotoxicity?

Given that this paper demonstrates a method of screening potential BiTE candidates, we believe that the Incucyte data shown is robust enough to show that screened BiTEs effectively activate T-cells against target cells and have no off-target effects. When viewing the supplementary videos provided, it is clearly shown that target cell killing begins as early as 24 hours, and is clearly shown to be an effective antigen-specific response. We feel that these videos and the accompanying data demonstrate t cell cytotoxicity without the need for radioactive isotopes or the like. Additionally, as this method removes the need for purification and quantification during screening, we feel that assays measuring tumor cell lysis such a chromium release assay would require the context provided by antibody quantification. 

It is not immediately clear what E:T ratios were used for experiments in Figure 5.

E:T ratios are included now included in the figure description of Figure 4 (previously Figure 5). This should clarify the data shown.

---

## [Decision Letter · Decision Letter 2]

29 May 2023

A Simplified Function-First Method for the Discovery and Optimization of Bispecific Immune Engaging Antibodies

PONE-D-22-22870R2

Dear Dr. Scott,

We’re pleased to inform you that your manuscript has been judged scientifically suitable for publication and will be formally accepted for publication once it meets all outstanding technical requirements.

Kind regards,

Masanori A. Murayama

Academic Editor

PLOS ONE

Additional Editor Comments (optional):

Congratulations. Your manuscript are endorsed at publish by reviewers.

Reviewers' comments:

Reviewer's Responses to Questions

**Comments to the Author**

1. If the authors have adequately addressed your comments raised in a previous round of review and you feel that this manuscript is now acceptable for publication, you may indicate that here to bypass the “Comments to the Author” section, enter your conflict of interest statement in the “Confidential to Editor” section, and submit your "Accept" recommendation.

Reviewer #1: All comments have been addressed

Reviewer #2: All comments have been addressed

2. Is the manuscript technically sound, and do the data support the conclusions?

Reviewer #1: Yes

Reviewer #2: Yes

3. Has the statistical analysis been performed appropriately and rigorously? 

Reviewer #1: Yes

Reviewer #2: N/A

4. Have the authors made all data underlying the findings in their manuscript fully available?

Reviewer #1: Yes

Reviewer #2: Yes

5. Is the manuscript presented in an intelligible fashion and written in standard English?

Reviewer #1: Yes

Reviewer #2: Yes

6. Review Comments to the Author

Reviewer #1: After careful examination of the revised manuscript, the response of the authors to previous reviews, and the changes made in the manuscript, I gather that the revised version of the manuscript has addressed the major concerns raised in the previous version of the paper (the limitations about unresolved comments are understandable). Overall, the article is now easy to read, and there is a logical interpretation of the data (with reasonable assumptions). Hence, I endorse the publication of this manuscript.

Reviewer #2: the authors have successfully addressed all the points and the manuscript is now of interest for the journal readership. It must be considered that BITE technology is a rapidly evolving area of investigation and the proposed approach is clearly of interest in the specific fiield

7. PLOS authors have the option to publish the peer review history of their article (what does this mean?). If published, this will include your full peer review and any attached files.

Reviewer #1: No

Reviewer #2: No

---

## [Editor Report · Acceptance letter]

13 Jun 2023

PONE-D-22-22870R2 

A Simplified Function-First Method for the Discovery and Optimization of Bispecific Immune Engaging Antibodies 

Dear Dr. McComb:

I'm pleased to inform you that your manuscript has been deemed suitable for publication in PLOS ONE. Congratulations! Your manuscript is now with our production department. 

Kind regards, 

on behalf of

Dr. Masanori A. Murayama 

Academic Editor

PLOS ONE